# Effect of Additives on Tribological Performance of Magnetorheological Fluids

**DOI:** 10.3390/mi15020270

**Published:** 2024-02-14

**Authors:** Songran Zhuang, Yongbing Cao, Wanli Song, Peng Zhang, Seung-Bok Choi

**Affiliations:** 1College of Information Science and Engineering, Northeastern University, Shenyang 110819, China; 20215409@stu.neu.edu.cn; 2School of Mechanical Engineering and Automation, Northeastern University, Shenyang 110819, China; 2370086@stu.neu.edu.cn; 3Nanjing Research Institute for Agricultural Mechanization, Ministry of Agriculture and Rural Affairs, Nanjing 210014, China; zhangpeng01@caas.cn; 4Department of Mechanical Engineering, The State University of New York, Korea (SUNY Korea), Incheon 21978, Republic of Korea; 5Department of Mechanical Engineering, Industrial University of Ho Chi Minh City (IUH), Ho Chi Minh City 70000, Vietnam

**Keywords:** magnetorheological fluid (MRF), tribology performance, additives, friction, wear surface

## Abstract

In this study, nano-diamond (ND) and MoS_2_ powder are used as additives in a carbonyl iron-based magnetorheological fluid (MRF) to improve its tribological performance. MRFs are prepared by dispersing 35 wt.% of CI particles in silicone oil and adding different proportions (0, 1, 3, or 5 wt.%) of ND and MoS_2_ additives. Seven kinds of MRFs are made and tested using reciprocating friction and wear tester under different normal loads, and then the friction characteristics are evaluated by analyzing the experimental results. The morphological properties of MRFs and contacting surfaces before and after the tests are also observed using a scanning electron microscope and analyzed via energy-dispersive X-ray spectroscopy. The results show that the appropriate weight percentage of MoS_2_ additives may decrease the friction coefficient and wear zone. It is also demonstrated from detailed analyses of worn surfaces that the wear mechanism is influenced not only by additives, but also by the applied normal load and magnetic field strength.

## 1. Introduction

As it is known that magnetorheological fluid (MRF) is a kind of smart suspension in a free-flowing liquid state in the absence of a magnetic field, under a magnetic field, its apparent viscosity can be increased by more than two orders of magnitude within milliseconds, and it exhibits solid-like characteristics displaying field-dependent yield stress [1,2]. MRF normally has three main constituents: magnetic nonlinear particles, nonmagnetic base liquids, and stabilizer additives. Micron-sized magnetizable particles, which play a key role in the effect of MR, are suspended in the base liquid or oil, and stabilizer additives are utilized to overcome sedimentation problems [3]. Since the discovery of the salient properties of MRF, various material components have been proposed and developed to improve the MR effect. Shah et al. [4] studied the influence of large-sized magnetic particles on the magneto-viscous properties of MRFs through comparing the rheological properties of three different types of MRFs in the presence of the magnetic field. The results showed that the addition of large-sized magnetic particles in magnetic fluid could increase the yield stress and the fluid stability under the field. Min et al. [5] examined the magnetic properties, material characteristics, and sedimentation properties of two different MRFs by dispersing both carbonyl iron (CI)/polyaniline and CI micro-spheres in silicone oil. The effect of surface treatment on the magnetic micro-spheres was discussed. Hato et al. [6] investigated the characterization of MRFs containing three different loadings of submicron-sized organoclay particles added into a CI suspension. And the dispersion stability of pure CI was improved with the increasing content of organoclay in the CI suspension. Han et al. [7] evaluated the effect of the MR particle corrosion on the performance of MRF and found that the shear stress controllability of the corroded MRFs was worse than that of the original fluid. Dong et al. [8] studied the effect of CoFe_2_O_4_ additives on the material performance and dispersion stability of CI-based MRFs under an external magnetic field. The results showed that CoFe_2_O_4_ additives improved the MR effects of the MRF, such as the shear stress, shear viscosity, and dynamic modulus.

Due to the many salient properties of MRFs, such as controllable yield stress and fast response time, numerous application systems have been proposed and studied so far in the fields of automotive industry, mechanical engineering, aerospace, construction, and healthcare [9]. However, before MRFs are applied, their durability and mechanical properties need to be sufficiently investigated. Recently, the effect of friction and wear has become recognized as a crucial factor that must be considered. Wong et al. [10] investigated the tribological behavior of an MRF without magnetic field activation using a block-on-ring tester. Seok et al. [11] examined the tribological properties of an MRF in a finishing process, and the results showed that the dominant wear mechanism in the finishing process was abrasion. In addition, a semi-empirical material removal model was proposed in this work for the description of the tribological behavior of the MRF. Wang et al. [12] studied the polishing effect of the fluid composition of the workspace of magnetorheological polishing (MRP), which consists of the concentration of carbonyl iron particles (CIPs), the concentration of cerium oxide (CeO_2_), and the size of cerium oxide (CeO_2_), using the reciprocating polishing method and by establishing regression equation models. Song et al. [13] proposed a simplified experiment with a disc-on-friction apparatus to investigate the frictional behavior of counter-faces lubricated by MRF under the magnetic field. Zhang et al. [14] studied the reciprocating friction characteristics of MRF for aluminum under different magnetic fields when MR fluids were worked with reciprocating motions. The experimental results showed that the average coefficient of friction under the magnetic field was lower than that without a magnetic field, regardless of loads and oscillation frequencies. Wang et al. [15] investigated the effect of the surface textures of specimens on the braking performance of an MR brake under different working conditions. On the other hand, several works on the tribological properties of MRFs have been carried out. Hu et al. [16] investigated the friction and wear properties of CI-based MRFs, which contained different constituent additives. It was found that MRFs showed controlled tribological properties under different magnetic fields. Reeves et al. [17] experimentally demonstrated that an appropriately sized boron nitride particle could enhance lubricity and minimize wear in the tribo-interface.

Until now, many materials, such as grease, iron naphthenate, lithium stearate, oleic acid, naphthenate, sulfonate, glycerol monooleate, silica, etc., have been used as additives. Additives can improve MRFs’ properties, such as the field-dependent yield stress and sedimentation of particles and interparticle friction, in addition to improving the mechanical properties of MRFs [3,18]. In recent decades, nano-diamonds (NDs) have been widely used as lubricants, greases, and coolants in the fields of machinery, metal processing, engine manufacturing, shipbuilding, aviation, transportation, etc. [19,20]. In general, ND additives added to oils or lubricants can improve the working lives of tools and reduce boundary friction and wear. For example, the friction coefficient was reduced to 20~30%; friction torque was reduced by 20~40%; and wear surfaces were reduced to 30~40% during the machining process [21]. However, as far as the authors are aware, there has been no report on the friction and lubrication characteristics of MRFs with ND additives. Consequently, the main technical contribution of this work is an experimental investigation of the tribological performance of CI-based MRFs with the additives of ND and MoS_2_ powder. To achieve the research goal, as a first step, seven types of MRFs were fabricated and investigated experimentally under different loadings and magnetic fields using a reciprocating friction and wear tester. Subsequently, the effect of the additive constituents and additive concentrations of MRFs on the friction and wear properties were observed. In addition, the surfaces of the specimens were examined using a surface profilometer, optical microscope, and scanning electron microscope (SEM) coupled with energy-dispersive X-ray spectroscopy (EDS) to investigate the field-dependent morphological characteristics.

## 2. Experimental Procedures

In this study, soft magnetic CI particles, silicone oil, stabilizers, and different proportion of ND and MoS_2_ additives were used as a dispersed phase, a suspending medium, and additives for MRFs, respectively. The specific volume fraction of MR fluid is presented in Table 1. The particle diameters of CI particles and additives were 2.5 μm and 4 μm, respectively. The CI MRF was prepared by dispersing 35 wt.% CI particles in the silicone oil; the CI/ND MRFs were prepared by adding 1, 3, or 5 wt.% ND to the CI MRF; and the CI/MoS_2_ MRFs were prepared by adding 1, 3, or 5 wt.% MoS_2_ to the CI MRF. The mixtures were treated using vibration dispersive mixing to ensure a uniform distribution of CI particles in the solution. Some methods were employed in the process, such as ultrasonic dispersion, high-speed dispersion, and mechanical milling [3].

The friction tests were performed using the reciprocating friction and wear tester (R&B 108-RF), and a schematic diagram of the experimental device is shown in Figure 1. The tester consisted of a motor, a crankshaft device, upper and lower holders, an electromagnet, and a loading system. In this device, the upper and lower holders were used to fix pin and disc samples, respectively. A direct current motor was used to provide accurate speed control to drive the pin samples with a periodic reciprocating motion on the disc samples using a crankshaft connecting rod system with a reciprocating stroke of 10 mm. An electromagnet, which was installed under the disc, provided sufficient magnetic field strength to the MRF. And the magnetic induction produced by the electromagnet was measured using a Tesla meter. It was found that a magnetic field strength of up to 25 mT could be applied to the surfaces of the samples. In this study, the appropriate magnetic field strength was applied at 10 mT. A computer control system was used to collect and process the experimental data from the transducers. A normal loading system was used for the external load, and the load cell was used to measure the friction force (*F*). And the friction coefficient was calculated using the following equation:(1)$$\mu_{k}=F/P$$
where *μ*_k_ is the kinetic friction coefficient; *F* is the nominal, measured friction force during sliding; and *P* is the applied load (normal force). Nominal friction force *F* was measured using the purchased module of friction signal analysis. The applied load *P* was loaded through a mechanical loading structure with a force display function. The samples for reciprocating the friction tests were aluminum. The diameters and lengths of the pin samples were 10 mm and 25 mm, the diameters and thicknesses of the disc samples were 60 mm and 8 mm, respectively. The disc samples were processed to be as thin as possible so as not to undermine the material mechanics performance. In order to reduce the effects of surface roughness on the experimental results as much as possible, the surface treatments of the pin and disc samples were carried out. In this study, all the tests were run under the same ambient conditions normally for the same duration of 1800 cycles. In order to understand the impact of variation on the friction coefficient, the oscillation frequency was set to 1 Hz and the load conditions to 5, 10, and 15 N. The specific friction conditions are presented in Table 2. During the tests, the friction coefficients on the contact surface were measured and recorded directly through an experimental apparatus. The pin and disc samples were cleaned using an ultrasonic cleaner before and after tests. The surfaces of the samples were examined using a surface profilometer and optical microscope. The CI particles were observed via SEM, and the chemical composition of the prepared MRF was analyzed using EDX.

## 3. Results and Discussion

### 3.1. Effect of Load and Magnetic Field

CI-based MRF with 3 wt.% ND additives was tested under the loads of 5, 10, and 15 N, and the experimental results are shown in Figure 2, Figure 3 and Figure 4. The kinetic friction coefficient changes along the cycle times, with and without the magnetic field, are shown in Figure 2. The results show that the curves mainly increased slightly with the test cycles. However, under the loads of 10 and 15 N, the curves increased rapidly along the first 500 cycles, and the kinetic friction coefficient showed a slight upward trend along the remaining cycle time. In addition, the kinetic friction coefficient increased with the decreasing load condition. The main reason for this is that there were peaks and valleys on the friction surface at the microscopic level; the kinetic friction force and load increased nonlinearly; and the micro contact area did not significantly increase under a larger normal load. Moreover, there were slight fluctuations in the contact between the pin and the disc during the wear process. The results also show that the kinetic friction coefficient under the magnetic field had higher values than the case without a magnetic field, but the tendency was similar in both cases. From the viewpoint of the rheological properties of MR fluid, the kinetic friction coefficient was mainly dominated by the chain formation of magnetic particles in local spatial positions between the friction pairs, and the increase in surface shear stress between friction pairs led to an increase in the kinetic friction coefficient. It is noted here that all experiments were repeated more than three times under the same conditions.

In order to further observe the surface morphology and wear changes, an optical microscope was used to evaluate the general morphology of the disc samples. Figure 3 shows optical microscopy images of the general morphology of the worn zone of the disc samples after the tests under different load and magnetic field conditions. Figure 3a,c,e show the abrasive positions of the disc samples without the magnetic field, and Figure 3b,d,f show the abrasive position of the disc with a magnetic field strength of 10 mT. The wear zone width and abrasion could also be seen on the disc surface after the tests. When a magnetic field strength of 10 mT was added, the wear zone width was obviously increased. But the tendency of the wear zone was different with and without the magnetic field.

Figure 4 shows the profile of the disc surface with CI-based MRF and 3 wt.% ND additives under different load and magnetic field conditions. The results show that the profile of the surface without the magnetic field had the lowest worn profile among all cases, and the magnetic field strength of 10 mT resulted in the largest worn profile. The width and depth of the worn profile increased with the increasing load conditions. During the test, the majority of magneto-rheological particles were trapped within the contact zone by the action of magnetic forces when a magnetic field was applied [22]. Due to the distribution of contact pressure between the pin and disc, the center position of the pin took the maximum contact pressure. The most wear occurred around 500 μm on the X-axis for all of the test cases when MR particles participated in the wear process.

### 3.2. Effect of Additive Constituents and Contents

Seven types of CI-based MRFs, with 0, 1, 3, and 5 wt.% contents of ND and MoS_2_ additives, were fabricated to investigate the effect of additives’ constituents and contents on the friction and wear properties of MRFs. The samples were tested under a load of 10 N with magnetic field strengths of 0 and 10 mT, and the results are shown in Figure 5, Figure 6, Figure 7, Figure 8, Figure 9 and Figure 10. The kinetic friction coefficient of MRFs with different contents of ND additives, along with the cycle times, are displayed in Figure 5. It can be seen that without the effect of a magnetic field, 3% of ND abrasive particles had a certain repairing effect on the surface during the initial friction stage, but the effect was not very good. However, 1% and 5% contents of ND abrasive particles had the opposite effect. Under the action of a magnetic field, the trend of the curve was almost similar to that without the action of a magnetic field. Moreover, the kinetic friction coefficient increased when the magnetic field strength increased from 0 to 10 mT, mainly due to the influence of magnetic particle chaining.

Figure 6 shows the kinetic friction coefficient of MRFs with different contents of MoS_2_ additives and different magnetic field strengths. It shows that the friction coefficient not only increased with the increasing MoS_2_ additive content, but also that the values were all lower than without additives. This indicates that MoS_2_ has a significant repairing effect on the friction surface, and the repairing effect of 1% content was obviously better than that of 3% and 5% contents. It was also found that the effect of magnetic field strength on the kinetic friction coefficients of MRFs with MoS_2_ additives is not obvious. Whether the magnetic field was applied or not, the MRFs with MoS_2_ additives had lower friction coefficients than that added ND additives. This means that the stability of the MRFs with MoS_2_ additives was better than those with ND additives. On the other hand, the friction coefficient of the base oil with ND particles presented irregular variation with the temperature [23,24,25].

Optical micrographs of the worn surfaces of disc samples after the tests are shown in Figure 7 and Figure 8. It can be observed that the worn areas on the disc surfaces not only increased with increasing contents of ND additives, but also increased with the applied magnetic field strength. For the MRFs with ND additives, the additives had a more obvious effect. But for the MRFs with MoS_2_ additives, the magnetic field strength had a more obvious effect. The density and viscosity of additives in MRF increased with the increasing additive content. The hardness and particle size of ND were greater than the CI particles of MRF, which led to aggravation of the wear and tear. When the magnetic field was applied, the CI particles were arranged in a chain structure, leading to an increase in the wear area. Comparing Figure 7 with Figure 8, it can be found that the width of the wear area with MoS_2_ additives was smaller than that with MRFs and ND additives. In addition, in order to compare the surface roughness of the disc with different additive contents after the experiment, a surface profilometer (SurfTest SV-3100, Mitutoyo, Kawasaki, Japan) was used to measure the arithmetical mean wear surface profiles of the discs. The measurement data were obtained at different loads and different additive contents, and the wear surface profiles are shown in Figure 9 and Figure 10. These figures show that the wear surface profiles increased with the increasing ND and MoS_2_ additive contents. And the best roughness was attained at an additive content of 1 wt.% for all the test data. It was also revealed that, for the MRFs with different content additives, the depth of the wear area increased with the additives content. And the depth of the wear areas of MRFs with MoS_2_ additives were smaller than those of the MRFs with ND additives. However, in Figure 10, it is shown that the depth of the wear area of the MRF with 1 wt.% MoS_2_ additives was smaller than that of the MRF without additives under 0 and 10 mT magnetic field strengths.

The SEM images and EDX results of the MRFs with ND or MoS_2_ additives before and after the tests are shown in Figure 11, Figure 12, Figure 13 and Figure 14. The particle sizes, the contents of ND and MoS_2_ additives, and MRF can be confirmed through these SEM images. Comparing the SEM images, it we confirmed that the sizes of CI particles changed, and were smaller after the test, and this was more obvious when the ND and MoS_2_ were added to MRFs. The EDX spectra showed the disc surface with MRFs and ND and MoS_2_ additives before and after the test, respectively. The changes in the chemical composition of the disc surface with MRFs and ND additives, with and without the magnetic field, after the test are shown in Figure 13. It can be seen that, after the tests, the contents of the elements C and Zr were increased. Because the influence of the magnetic field increased the wear and tear, the contents of Fe and C elements increased more when the magnetic field was applied after the test. Figure 14 shows the chemical composition of the disc surface with MR fluids and MoS_2_ additives before and after the test. The chemical composition data show that the C and Fe contents were reduced, and the Mo, S, and Fe contents were increased. Overall, the involvement of iron particles and other components from the MRF led to a change in the chemical composition of the plate surface after the test, thus causing the Al content to increase on the coating surface after the friction wear test.

## 4. Conclusions

In this study, the friction and wear characteristics of CI-based MRFs with ND and MoS_2_ additives (1, 3, and 5 wt.%) were investigated under loads of 5, 10, and 15 N with 1800 cycles, respectively. The results show that the coefficient of friction increased as the load condition decreased. The width and depth of the wear area increased with the increasing weight percentages of ND and MoS_2_ additives. Compared with the MRF without additives, the width and depth of the wear area decreased when 1 wt.% MoS_2_ additives were added to MRF. The density and viscosity of additives in the MRFs also increased with the increasing weight percentage of the additives, which may have led to the increment in the width and depth of the wear. When the magnetic field was applied, the friction coefficient of the MRFs decreased. On the contrary, the wear zone’s width and depth increased when the magnetic field was applied. The width and depth of the wear area with MoS_2_ additives was smaller than that with ND additives. In addition to the contents of Fe and C elements, the contents of Zr, Mo, and S elements were also increased after the tests when ND and MoS_2_ additives were added to MRFs. It has been shown in this work that the friction characteristics of MRFs can be improved through the addition of appropriate additives, resulting in the positive lubrication performance of many practical application systems involving MRFs.

## Figures and Tables

**Figure 1 micromachines-15-00270-f001:**
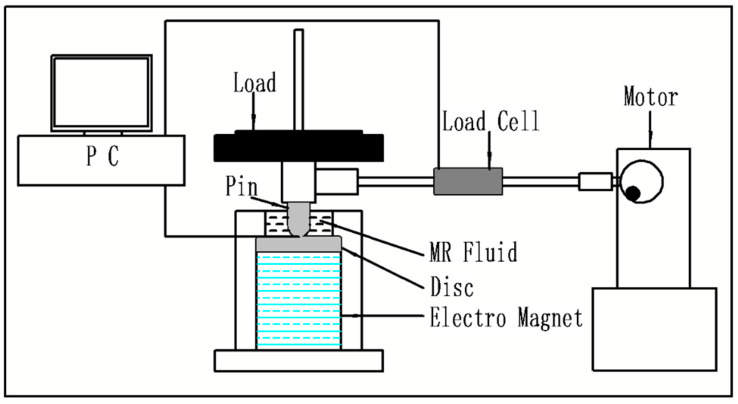
Schematic diagram of the reciprocating friction and wear tester.

**Figure 2 micromachines-15-00270-f002:**
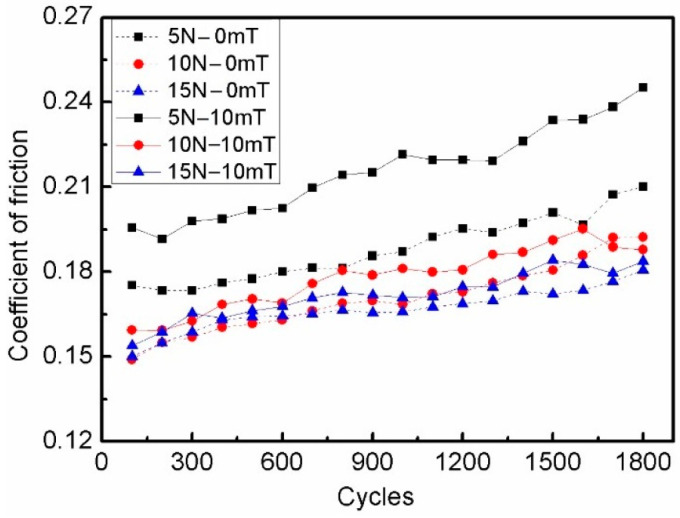
Kinetic friction coefficient of MRF with 3 wt.% ND additives under different loading conditions.

**Figure 3 micromachines-15-00270-f003:**
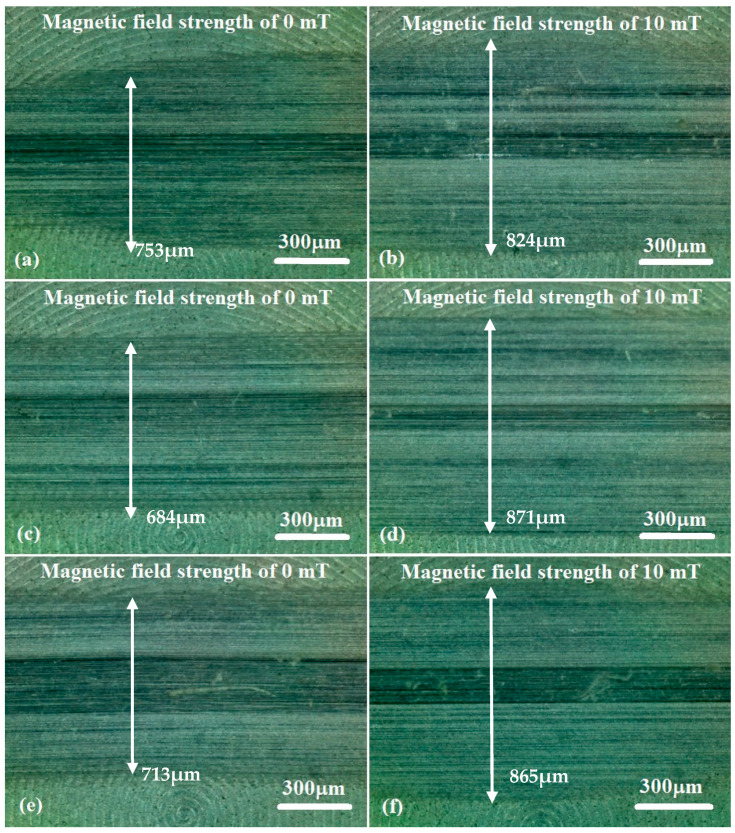
Optical microscope micrographs of disc samples after the tests with MRF and 3 wt.% ND additives under different load conditions: (**a**,**b**) 5 N; (**c**,**d**) 10 N; (**e**,**f**) 15 N.

**Figure 4 micromachines-15-00270-f004:**
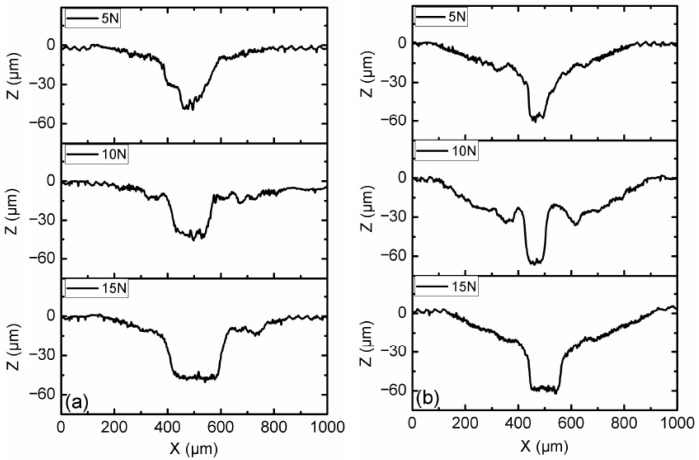
Wear surface profiles of disc samples with different load conditions: (**a**) magnetic field strength of 0 mT; (**b**) magnetic field strength of 10 mT.

**Figure 5 micromachines-15-00270-f005:**
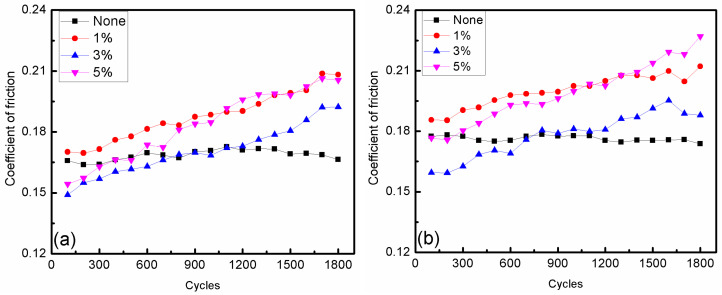
Change in the kinetic friction coefficient with different ND additive contents: (**a**) magnetic field strength of 0 mT; (**b**) magnetic field strength of 10 mT.

**Figure 6 micromachines-15-00270-f006:**
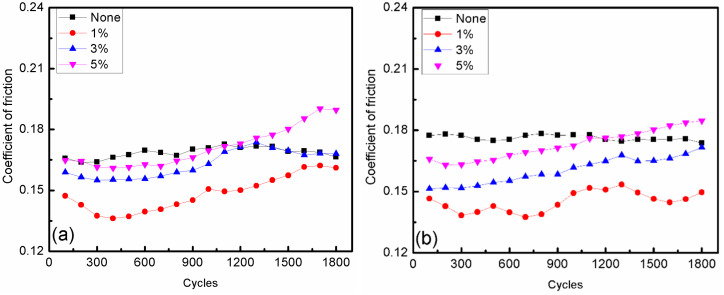
Change in the kinetic friction coefficient with different MoS_2_ additive contents: (**a**) magnetic field strength of 0 mT; (**b**) magnetic field strength of 10 mT.

**Figure 7 micromachines-15-00270-f007:**
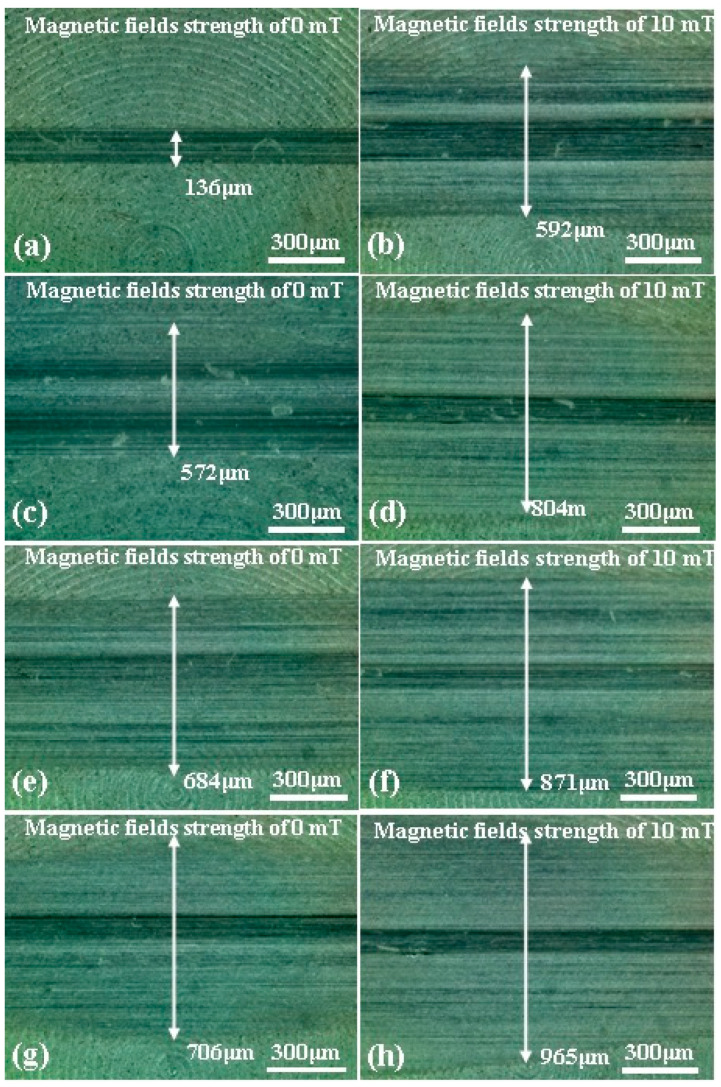
Optical microscope micrographs of discs after the tests using MRFs with different ND additive contents: (**a**,**b**) 0%, (**c**,**d**) 1 wt.%, (**e**,**f**) 3 wt.%, (**g**,**h**) 5 wt.%.

**Figure 8 micromachines-15-00270-f008:**
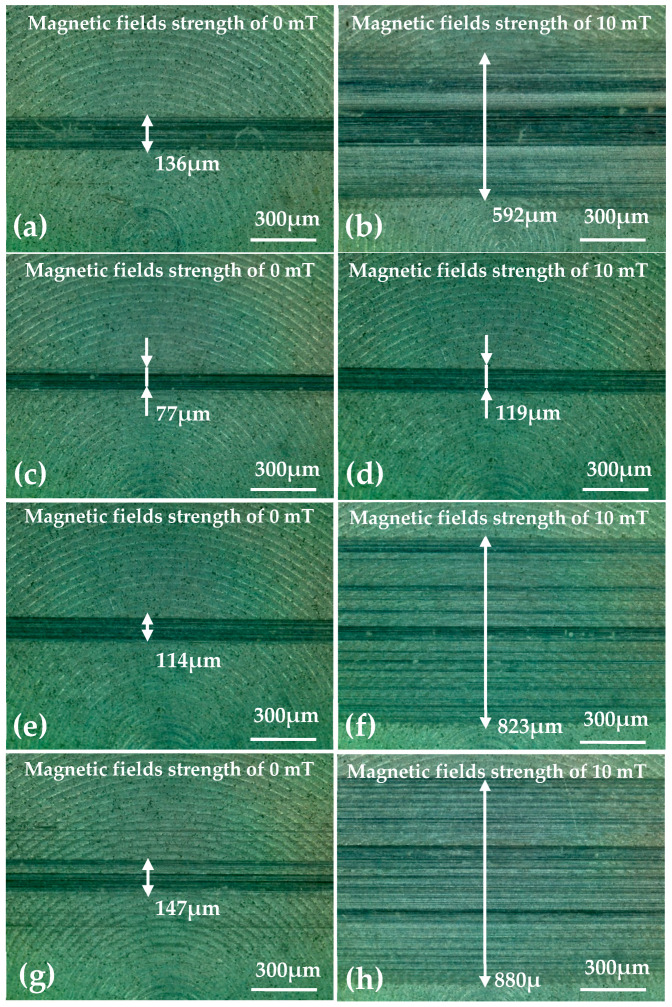
Optical microscope micrographs of discs after the tests using MRFs with different MoS_2_ additive contents: (**a**,**b**) 0%, (**c**,**d**) 1 wt.%, (**e**,**f**) 3 wt.%, (**g**,**h**) 5 wt.%.

**Figure 9 micromachines-15-00270-f009:**
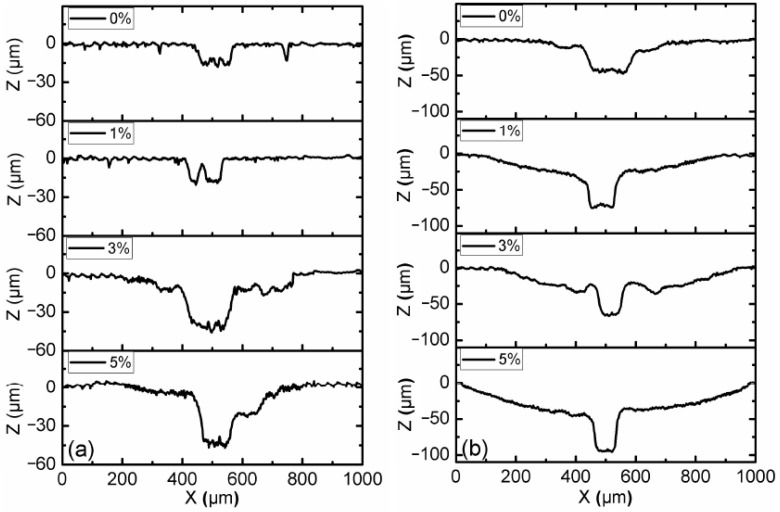
Wear surface profiles of discs with different ND additive contents: (**a**) magnetic field strength of 0 mT; (**b**) magnetic field strength of 10 mT.

**Figure 10 micromachines-15-00270-f010:**
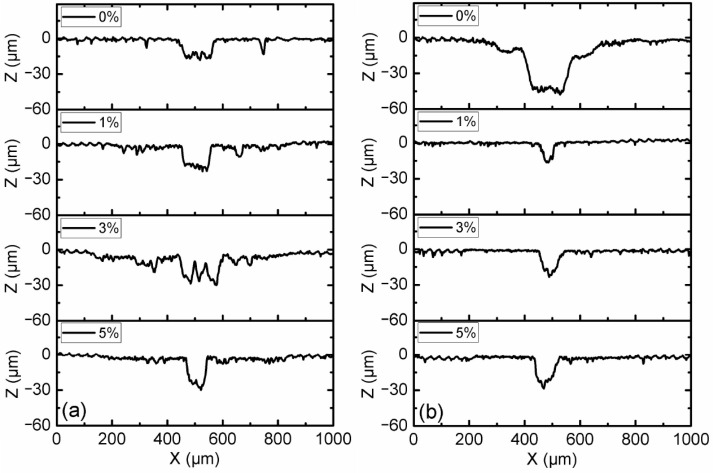
Wear surface profiles of discs with different MoS_2_ additive contents: (**a**) magnetic field strength of 0 mT; (**b**) magnetic field strength of 10 mT.

**Figure 11 micromachines-15-00270-f011:**
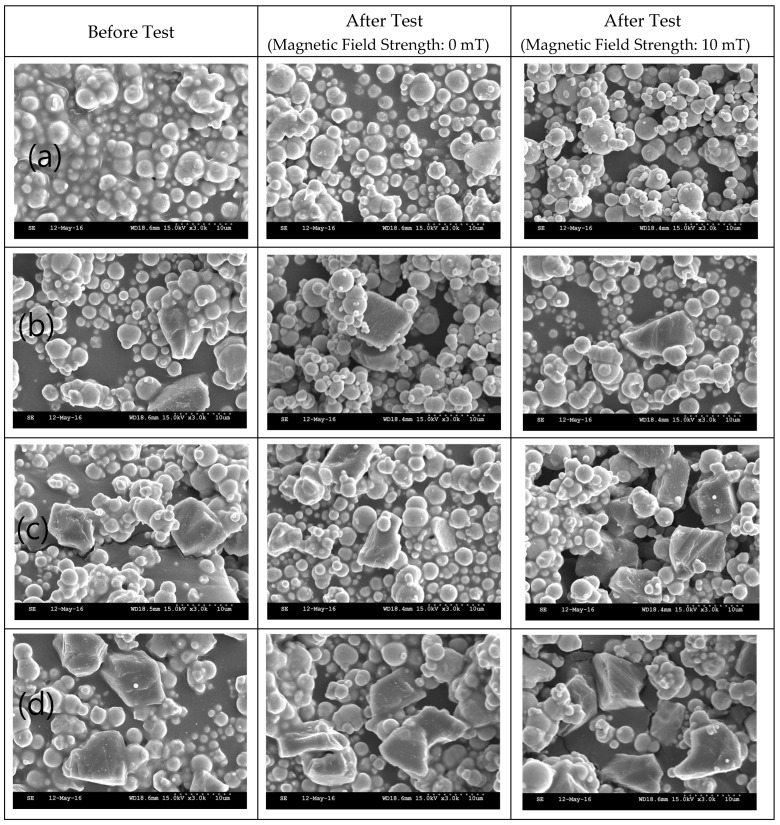
SEM of MRFs with different ND additive contents before and after tests: (**a**) 0, (**b**) 1, (**c**) 3, (**d**) 5 wt.%.

**Figure 12 micromachines-15-00270-f012:**
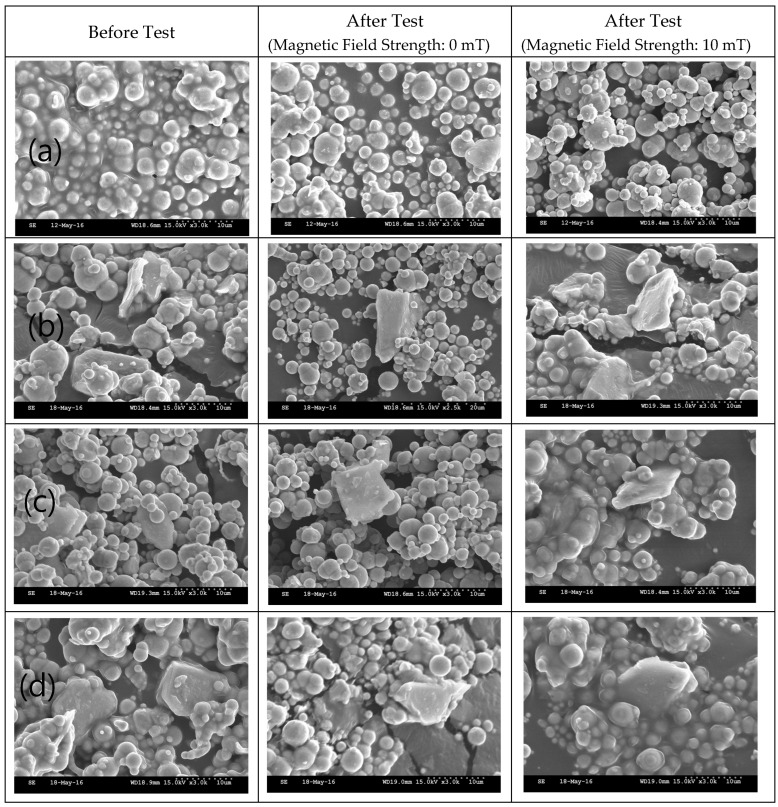
SEM of MRFs with different MoS_2_ additive contents before and after tests: (**a**) 0, (**b**) 1, (**c**) 3, (**d**) 5 wt.%.

**Figure 13 micromachines-15-00270-f013:**
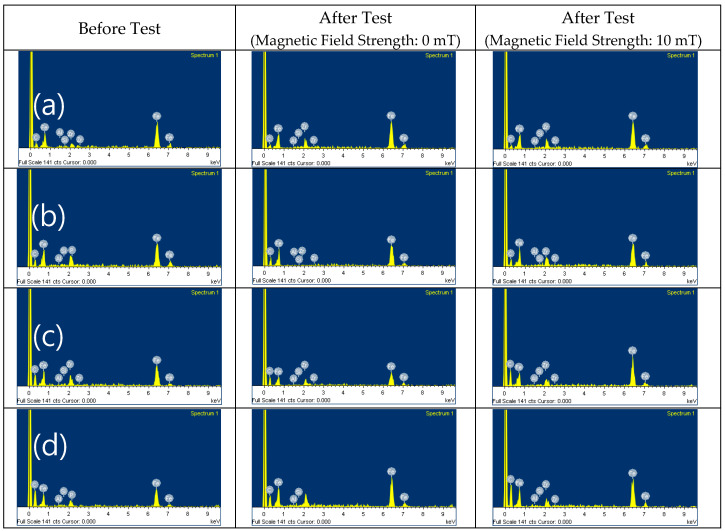
EDX of MRFs with ND additives: (**a**) 0, (**b**) 1, (**c**) 3, (**d**) 5 wt.%.

**Figure 14 micromachines-15-00270-f014:**
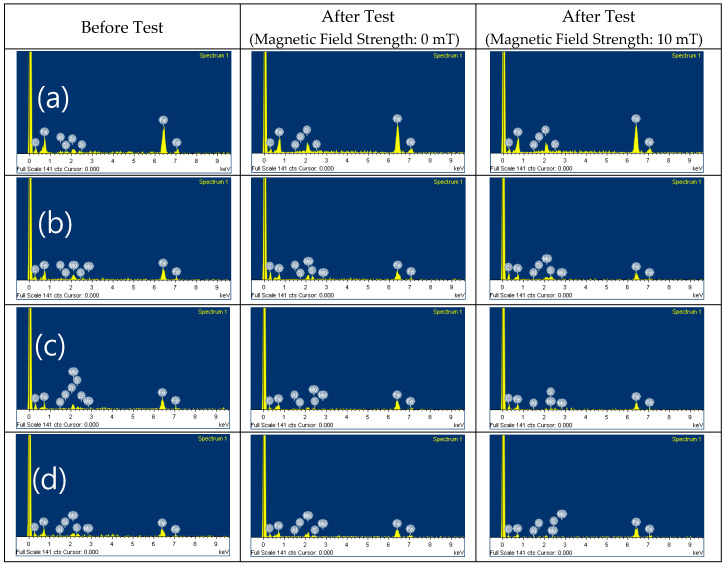
EDX of MR fluids with MoS2 additives: (**a**) 0, (**b**) 1, (**c**) 3, (**d**) 5 wt.%.

**Table 1 micromachines-15-00270-t001:** The specific volume fraction of MR fluid.

Parameter	Content of Component (wt.%)
Silicone oil	55–59
CI particles	35
Sodium dodecyl sulfate	3
Sodium nitrate	1
Nanodiamond/MoS_2_	1–5
Glycerin	1

**Table 2 micromachines-15-00270-t002:** The specific friction conditions.

Parameter	Values
Magnetic field (mT)	0, 10
Load (N)	5, 10, 15
Oscillation frequency (Hz)	1
Reciprocating stroke (mm)	10
Temperature (°C)	25
Cycle	1800

## Data Availability

The raw data supporting the conclusions of this article will be made available by the authors on request.

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
