# Peer review of "Effect of Additives on Tribological Performance of Magnetorheological Fluids"

_micromachines, 2024, doi:10.3390/mi15020270_

Round 1

Reviewer 1 Report

Comments and Suggestions for Authors

Reviewer 2 Report

Comments and Suggestions for Authors

The authors presented original studies of the friction and wear characteristics of CI-based MRFs with ND and 270 MoS2 additives (1, 3 and 5 wt.%) were investigated under the load of 5, 10 and 15N, 1800 271 cycles, respectively.

The authors developed an original installation, which they used to study the coefficient of friction of magnetorheological fluids with various additives. The authors investigated changes in the structure of the sample and the properties of colloids. The necessary theoretical interpretation is given.

Several comments can be made on the article:

1. The article does not contain information on the colloidal stability of magnetorheological fluids with the studied additives. The article makes no mention of any surfactants that were coated on the nanoparticles.

2. The experimental setup must be described in more detail, indicating the names of the devices and sensors used, as well as the characteristics of the electromagnet and drives.

3. None of the experimental graphs indicate the measurement error

4. The authors use only one field value of 10 mT. The article would look more presentable if the dependence of the friction coefficient on different values of magnetic field induction was given

5. In conclusion, the authors state “When the magnetic field is applied, the friction coefficient of MRFs is decreased.” This contradicts the graphs presented in Figure 2,5,6.

After these minor changes, the article can be published.

Round 2

Reviewer 1 Report

Comments and Suggestions for Authors

Comments 6: What is the depth of immersion of the pin in the MR fluid? Isn't the increase in the kinetic friction coefficient caused by the increase in the resistance force of the MR liquid to the moving pin after the increase in its apparent viscosity?

The authors explained that the depth of immersion of the pin in the MR fluid is approximately 10 mm and does not affect the nominal friction force F is measured by the purchased module of friction signal analysis. I suggest an experiment by immersing the pin to a depth of 5 mm or less and taking the measurement again. In this way, the lack of relationship between the force and coefficient of friction and the resistance of the liquid due to the change in its apparent viscosity will be demonstrated.

This is a proposal for further research - beyond this study.